# Neural representation of words within phrases: Temporal evolution of color-adjectives and object-nouns during simple composition

Maryam Honari-Jahromi[1]ᐤ, Brea Chouinard[2]ᐤ, Esti Blanco-Elorrieta[3,4], Liina Pylkkänen[3,4,5], Alona Fyshe[2,6,7]*

1 Department of Computer Science, University of Victoria, Victoria, BC, Canada, 2 Department of Computing Science, University of Alberta, Edmonton, AB, Canada, 3 NYUAD Institute, New York University, Abu Dhabi, UAE, 4 Department of Psychology, New York University, New York, NY, United States of America, 5 Department of Linguistics, New York University, New York, NY, United States of America, 6 Department of Psychology, University of Alberta, Edmonton, AB, Canada, 7 Alberta Machine Intelligence Institute, Edmonton, AB, Canada

ᐤ These authors contributed equally to this work.
* alona@ualberta.ca

**Data Availability Statement:** The data underlying this study is available on OSF (https://osf.io/p7gc6/

## Abstract

In language, stored semantic representations of lexical items combine into an infinitude of complex expressions. While the neuroscience of composition has begun to mature, we do not yet understand how the stored representations evolve and morph during composition. New decoding techniques allow us to crack open this very hard question: we can train a model to recognize a representation in one context or time-point and assess its accuracy in another. We combined the decoding approach with magnetoencephalography recorded during a picture naming task to investigate the temporal evolution of noun and adjective representations during speech planning. We tracked semantic representations as they combined into simple two-word phrases, using single words and two-word lists as non-combinatory controls. We found that nouns were generally more decodable than adjectives, suggesting that noun representations were stronger and/or more consistent across trials than those of adjectives. When training and testing across contexts and times, the representations of isolated nouns were recoverable when those nouns were embedded in phrases, but not so if they were embedded in lists. Adjective representations did not show a similar consistency across isolated and phrasal contexts. Noun representations in phrases also sustained over time in a way that was not observed for any other pairing of word class and context. These findings offer a new window into the temporal evolution and context sensitivity of word representations during composition, revealing a clear asymmetry between adjectives and nouns. The impact of phrasal contexts on the decodability of nouns may be due to the nouns' status as head of phrase—an intriguing hypothesis for future research.

) and the code is available on GitHub (https://github.com/fyshelab/NeuralPhraseComposition).

**Funding:** This research was supported by the NYUAD Research Institute (https://nyuad.nyu.edu/en/) under Grant G1001 (LP), by the Natural Sciences and Engineering Research Council of Canada (NSERC, https://www.nserc-crsng.gc.ca/index_eng.asp) through a Discovery Grant (AF), and the Canada CIFAR (Canadian Institute for Advanced Research, https://www.cifar.ca/) AI Chair program (AF). The computational work was supported in part by infrastructure made available by WestGrid (https://www.westgrid.ca/) and Compute Canada (https://www.computecanada.ca/) (AF). These funders played no role in the study design, data collection and analysis, decision to publish, nor preparation of the manuscript.

**Competing interests:** The authors have declared that no competing interests exist.

## Introduction

What is the relationship between the neural representation of a single word, occurring in isolation, and the representation of that same word in a combinatory context? In natural language, context can morph word meanings in many ways. One of the most obvious ways is via disambiguation: for example, in the phrase 'term paper,' 'term' means *semester* and 'paper' means '*piece of writing*' even though both of these words have many other uses as well. Thus, the neural representations of 'term' and 'paper' may differ robustly depending on the context.

This study uses a decoding approach to address the general question of how combinatory contexts affect the neural representations of word meanings. We see value in starting with relatively straightforward cases, and thus did not investigate ambiguous cases such as those just mentioned. Instead, our combinatory contexts were all noun phrases comprised of a color-adjective and an object-describing noun, such as 'blue cup.' This type of composition has been studied extensively with time-sensitive magnetoencephalography (MEG), with results implicating the left anterior temporal lobe and the ventromedial prefrontal cortex as relatively stable correlates of composition [1].

Notably, unlike prior decoding work that investigated semantic representations during *comprehension* (e.g., [2]), the current MEG study involved language *production* to measure the planning of words millisecond by millisecond, during a picture naming task. Pictures of colored objects (e.g., white lamp) were named either with single nouns ("lamp"), single adjectives ("white"), or with combinations of those adjectives and nouns ("white lamp"). All pictures also contained a background color, allowing for an additional 'list' control, where participants named the background color plus the object shape (e.g., "green", "lamp"). This created two-word utterances that do not form a conceptual combination of the sort that noun-adjective pairs create in natural language. This 'list' condition fully controlled for the number of uttered words and their lexical characteristics, allowing us to examine just the role of composition.

From prior MEG work on the planning of adjective-noun phrases in picture naming, we know that activity increases reflecting composition can be observed as early as 100-200ms after picture onset [3–5]. Estimates of the timing of lexical access in production fall into a similar time window: at around 150ms, both phonological and semantic properties of words are activated, as measured by MEG [6]. This suggests that lexical access and composition proceed in parallel. Against this background, context effects on lexical representations could start as early as 100-200ms in our study.

We used computational models of language (i.e. a word embedding model; [7]) and machine learning algorithms to detect semantic representations in the brain. We will use the term *decodability* to refer to the ability of our computational models to detect the semantic information related to the stimulus using a recording of brain activity. Decodability has been explored for people reading words in isolation [8, 9] phrases in isolation [2], sentences [10] and stories [11]. Decoding has also been successful using data collected while people listen to language [12, 13]. Our work is, to our knowledge, the first to use decoding techniques and MEG to detect the semantics of words before they are uttered.

Our design allowed us to quantify the relative decodability of noun and adjective neural representations in isolation, and when combined into adjective-noun lists or adjective-noun phrases. Classic syntactic theories hold that the features of the head of a phrase are inherited by the entire phrasal node [14], which in neural terms could mean that the representation of a head is stronger, and thus more decodable, than the representation of a modifier. In our study, this would result in the noun, as the head of the phrase, having greater decodability than the adjective. In contrast, theories in formal semantics posit that the intersective modification of nouns by adjectives proceeds via a fully symmetric predicate modification rule [15]. Thus, in

our study, equal decodability of both the adjective and the noun within the phrases could be interpreted as a reflection of this type of symmetric semantic composition.

Additionally, decoding from a time-sensitive measurement like MEG allowed us to address neural representations across time. These analyses can be done in three ways. First we can train our model and then test using held-out data, where both train and test data come from the same time window (same-window-decoding). This typical decoding analysis allows us to measure the robustness of a neural signature within a time window. Second, we can also keep the time window constant, but choose held-out test data from *another condition*, testing how similar a representation is across conditions (across-condition). Third, we can choose *a different time window* from which to draw our held-out test data, testing the robustness of the pattern in time (resulting in a temporal generalization matrix, or TGM). This third analysis type allows us to test if a semantic representation is held constant in the brain over some time period, or if it re-emerges at different time points. Classic language production models propose a sequence of activated representations, proceeding from concept to lexeme to phonological representation and so forth [16] but say nothing about whether, for example, the conceptual representation stays active past the initial processing stages. Our data allowed us to ask whether representations detected by our models at a certain time after picture onset were also decodable at later times, or whether the processing stream only showed the characteristics of the classic models, where neural representations transform into new representations as we move towards articulation.

To summarize, we ask to what extent the neural representation of a word is the same when it is prepared for production as a single word compared to when it is prepared as part of a meaningful phrase. We do this using computational models of language meaning to compute decodability for adjectives and nouns when they are in isolation, and when they are part of a combinatory phrase vs. a non-combinatory list. We also evaluate the persistence and/or re-emergence of a word's meaning, in isolation and in phrases and lists, using TGMs.

## Methods

All experimental protocols were approved by New York University Institutional Review Board and conducted in accordance with the relevant guidelines and regulations, and all participants signed an informed consent form before taking part in the experiment. The data were originally collected to investigate the relationship between spoken and signed language with regards to the neural correlates of basic composition in language production [4]. In the current study, we used only the spoken language data. Our goal was to compare pre-utterance semantic representations in non-compositional versus compositional/phrasal contexts.

### Participants

Nineteen right-handed monolingual native English speakers (9 female; ages: Mean: 25.6, 95 SD = 7.3), all neurologically intact with normal or corrected to normal vision, provided their written consent to participate in the original study [4].

### Stimuli

Participants named pictures that depicted a colored object (e.g., white lamp) on a colored background (e.g., green) (Fig 1). All conditions used the same stimuli, and instructions at the beginning of the block differentiated the naming task to be performed in each condition. Non-compositional utterances were elicited by asking the participant to (i) say the name of the object color (i.e., "white"; adjective-only context); (ii) say the name of the object (e.g., "lamp"; noun-only context); or (iii) to say the background color, then pause, then the object name in a list-like fashion (e.g., "green", "lamp"; list context). In the compositional context, participants

## a) Experimental conditions

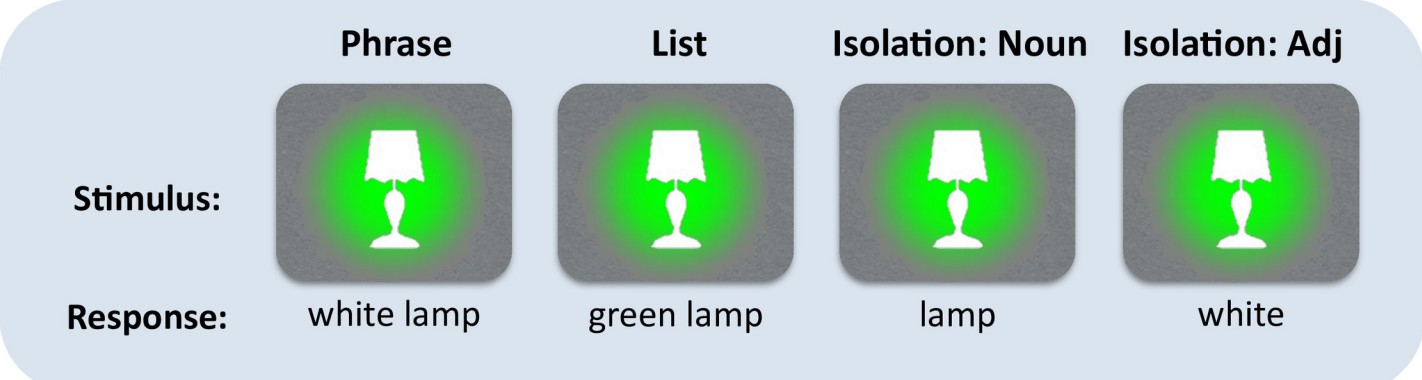

## b) Trial design: Name the colored object on the screen

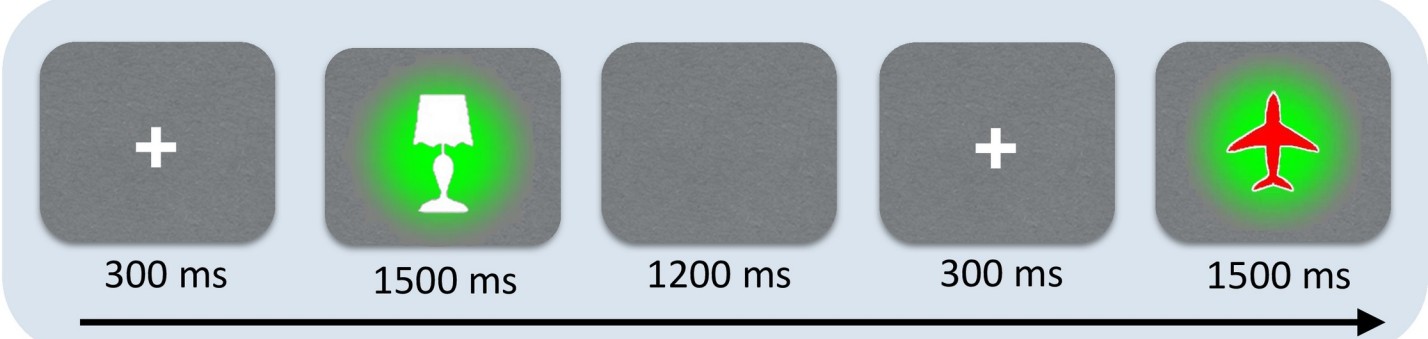

**Fig 1. Experimental design and trials structure.** Participants named colored objects in three ways, depending on task instruction: as phrases (white lamp), as single nouns (lamp), as single adjectives (white) or as adjective-noun lists (green, lamp), naming the color of the background followed by the object name. Our analyses assessed the decodability of adjective and/or noun representations in these four contexts.

were directed to describe the colored object on the screen by saying its color followed by its shape (e.g., "white lamp"; phrase context). Examples of images and utterances for each context appear in Fig 1A. There was also a control condition wherein the participant was instructed to say the background and object colors, but it was not analyzed here. The order of blocks was randomized across participants with the only constraint being that two blocks of the same condition never appeared consecutively. We controlled for frequency across word types (adjectives vs. nouns) using frequencies was extracted from Balota et al. [17]. Average noun frequency was 14396, average adjective frequency was 14976 (t = -0.47, p = .640742).

In total, the experiment consisted of 500 trials in which participants viewed one of 25 unique images created from a subset of five object shapes (bag, bell, cane, lamp, plane) and five colors (black, brown, green, red, white). Each image was also given a background color, which was counterbalanced so that background colors appeared an equal number of times, equally distributed across the different shapes. We also swapped the object and background colors to create a complementary set of 25 stimuli, thus creating 50 stimuli images in total. These 50 items were presented twice each for a total of 100 trials per condition. The items were presented in blocks of 25, for a total of 4 blocks per condition.

## MEG procedure and preprocessing

Prior to the MEG recording, each participant's head shape was digitized by a Polhemus dual source hand-held FastSCAN laser scanner (Polhemus, 112 VT, USA). The MEG data were recorded using a 208-channel axial gradiometer system (Kanazawa Institute of Technology, Kanazawa, Japan) at Neuroscience of Language Lab in NYU Abu Dhabi. MEG data were collected with a sampling frequency of 1000Hz (200 Hz low-passed filter). An MEG compatible microphone (Shure PG 81, Shure Europe GmbH) was used to record uttered speech of the participants. Each trial started with a fixation cross for 300ms, followed by the stimulus image, which was present until participant's response or timeout (1500ms, see Fig 1B). Afterwards, a break of 1200ms was given until appearance of the fixation cross belonging to the next trial. Trials were epoched at 100ms before to 700ms after stimuli onset to avoid contamination via motion artifact coinciding with overt speech and noise was reduced using Continuously Adjusted Least-Squares Method [18]. Epochs were baseline corrected using the average of a 100ms interval prior to the stimulus onset. Unlike Blanco-Elorrieta et al. [4], we rejected only those trials with erroneous responses. MEG signals were band-passed using a Butterworth filter of order 20 between 0.1Hz and 40Hz. We used no ICA artefact rejection or blink / heart beat removal, as such artefacts are less problematic in decoding studies.

Our analysis operates on the within-participant average over trials of a particular word in a particular context. Based on the context and word-type of interest, we first chose a target word and then select trials with a target utterance containing that word. Within this set of selected trials, we averaged random groups of 5 epochs to minimize noise in the signal. This yielded 4 averaged epochs per noun and 20 averaged epochs in total. We followed the same procedure of averaging epochs for the adjectives. Trials were averaged within participant; we trained separate models for each participant, and report the average model performance. Further decoding analysis and statistical significance tests were conducted in Scikit-learn [19] and MNE-Python [20] and FieldTrip [21]. The code for all analysis is available at https://github.com/mahon94/compositionInBrain.

## Decoding neural signatures using computational models of language

We calculated decoding accuracies to determine if the computer model could detect the semantic properties of the words to be uttered from the MEG data. Here, our computer model consists of word embeddings that represent the semantics of single words, and a regression model to map the MEG data to the dimensions of the word embeddings. We used Skip-gram word embeddings [22], which are derived from a neural network model trained on the Google News dataset (an internal Google dataset with one billion words) to predict context words. We were interested in how decoding accuracy varies over time, so we trained using 100ms windows of time, shifted in increments of 5ms across the full time window (i.e., -100 to +700ms relative to stimulus onset). An overview of the data organization, training procedure and 2 vs. 2 test appears in Fig 2.

Similar to the approach described in Fyshe et al. [2], we form the MEG dataset $X \in \mathbb{R}^{N \times p}$ where $N = 20$ is the total number of averaged epochs reshaped to vectors of size $p = c \times t$, for $c = 208$ MEG gradiometer sensors with $t = 100$ time samples per window of analysis. We normalize each column of $X$ to have mean 0 and standard deviation 1, and append a column of ones to account for the bias term. We then train d independent L2-regularized (ridge) regression models $h_j(X), j \in \{1,2,\ldots,d\}$ to predict each column of the matrix of word embeddings $Y \in \mathbb{R}^{N \times d}$ ($d = 300$ is dimension of the Skip-gram vector). The $j$th regression model is trained as

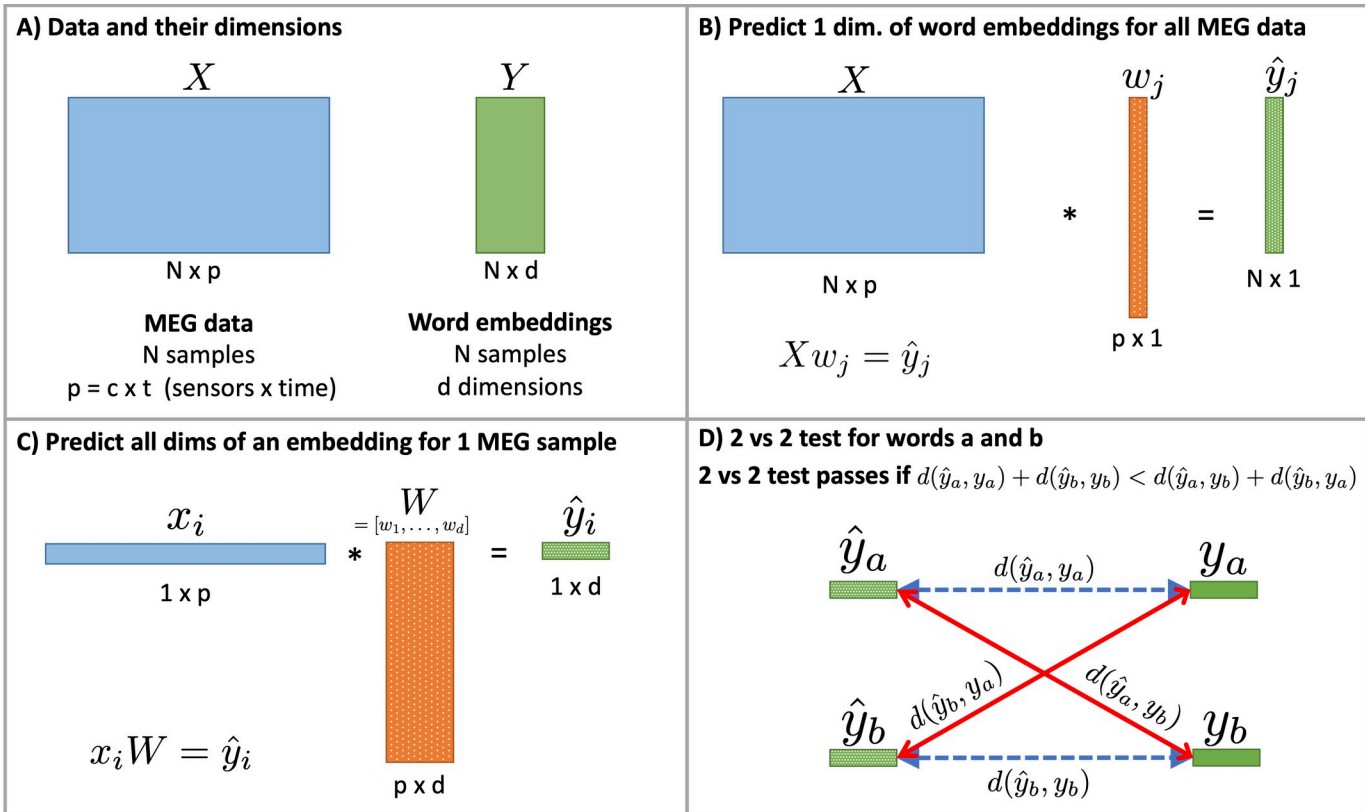

**Fig 2. Explanation of data, model, and testing procedures.** Values that are fixed appear as solid-colored rectangles, and values that are learned or predicted appear as dotted rectangles. A) The dimensions of the MEG data X (blue) and the word embedding matrix Y (green). B) The process for predicting one dimension (j) of the word embedding matrix Y. Note that this is corresponds to $h_j(X)$ in the in-text equations. C) Predicting all dimensions of a word embedding for MEG data sample $x_i$. W is the concatenation of w vectors from B). D) The 2 vs 2 test. The 2 vs. 2 test measures how similar the predictions $(\hat{y}_a, \hat{y}_b)$ are to their corresponding ground truth vectors $(y_a, y_b)$ using a vector distance criterion d(v,u). If the correct matching of true to predicted vectors (blue lines) represents a smaller distance than the incorrect matching (red lines), the 2 vs 2 test passes.

follows:

$$h_j(X) = Xw_j,$$

$$w_j = \underset{w}{\mathrm{argmin}} \, \|Xw - y_j\|_2^2 + \lambda w^T w$$

$$= (X^T X + \lambda I)^{-1} X^T y_j$$

Where $w_j \in \mathbb{R}^p$, $\|A\|_2^2$ indicates the squared two norm of A, and $y_j$ is the jth column of matrix Y. We determine the best performing regularization parameter $\lambda$ separately for each column of the word embedding matrix using leave-one-out cross validation. Since $N \ll p$, we speed up training using the kernel trick based on singular-value decomposition (SVD) demonstrated by Hastie & Tibshirani [23], and regularization helps to control overfitting. Note that every model was trained separately for each participant. Thus, the patterns underlying the decoding accuracies we observed may not be stable across people, and our analysis did not test for such stability. Rather, we tested for the presence of a pattern, and if the patterns generalize across time and condition within a participant's data. Our methodology then tests if the average decoding

accuracy (a function of the participant-specific patterns) shows stable patterns across participants.

As mentioned previously, the regression model can be trained and then tested within or across conditions. For simplicity, we will refer to each train/test regime using a pair of context names separated by a slash, with the word before the slash referring to the training context, and the word after the slash referring to the testing context. For example, if we were to both train and test in isolation, we would refer to it as isolation/isolation. If we were to train in the isolation context and test in the phrase context, we would refer to it as isolation/phrase.

Within-context accuracy (e.g., phrase/phrase) indicates the consistency of the representation across trials within a context. Across-context accuracy (e.g., isolation/phrase) indicates how consistent the neural representation is between the two contexts. Thus, high accuracy for nouns in a isolation/phrase analysis would indicate that the neural representation of the noun is similar in isolation and in list context. Accuracy was computed using the 2 versus 2 test.

## Computing decoding accuracy using the 2 versus 2 test

On a dataset of N averaged epochs, we hold out 2 averaged epochs and the two corresponding target vectors ($y_i$, $y_j$). We train the model on the remaining N-2 averaged epochs and N-2 target vectors. Testing the model on the 2 held-out averaged epochs provides two predicted semantic vectors ($\hat{y}_i$, $\hat{y}_j$). The 2 vs. 2 test measures how similar the predictions ($\hat{y}_i$, $\hat{y}_j$) are to their corresponding ground truth vectors ($y_i$, $y_j$) using a vector distance criterion d(v,u). While any kind of distance metric can be used, we opt for cosine distance. In particular, the test passes if the following equation holds:

$$d(\hat{y}_i, y_i) + d(\hat{y}_j, y_j) < d(\hat{y}_i, y_j) + d(\hat{y}_j, y_i) \tag{1}$$

where the distance of matching vectors is smaller than the distance of non-matching vectors. We award a score of 1 if the test passes, 0 if it fails, and 0.5 if the two summations are equal. The reported 2-vs-2 accuracy is the average score of the 2-vs-2 test on every possible pair in the dataset (20 choose 2, denoted $\binom{20}{2} = 190$). Recall that our 20 data instances contain 4 samples for each word. For this reason, 30 of the 190 2-vs-2 pairs will pair two samples of the same word. Such a pairing renders the 2-vs-2 test degenerate because $y_i = y_j$ and the two halves of Eq (1) are equal. Thus, there are a total of $\binom{20}{2} - 30 = 160$ valid 2-vs-2 pairs. An illustration of the 2 vs. 2 test appears in Fig 2D.

**Statistical significance.**    To assess the statistical significance of the 2-vs-2 accuracy, we use permutation tests. In permutation tests we randomly shuffle the mapping of target utterances to MEG epochs. This simulates the scenario where there is no meaningful relation between the MEG recordings and the target utterances. We train and test our model on datasets built with 100 randomly shuffled utterance-to-epoch mappings, producing 100 decoding accuracies. As expected, the mean decoding accuracy on the shuffled datasets is at chance (50%). We then fit a normal kernel density function to the histogram of decoding accuracies to form a null distribution. From this null distribution, we calculate the *p*-value for the decoding accuracy of models trained with the original un-permuted labels. To determine if results are above chance, we correct the *p*-values for multiple comparisons over time using False Discovery Rate (FDR) with no dependency assumption (Benjamini-Hochberg-Yekutieli method; [24]).

When comparing the effect of context and word category on accuracy of the models, we find clusters of time where the 2-vs-2 accuracy differs significantly using a 2 x 3 ANOVA combined with the cluster permutation method [25]. We submit the 2-vs-2 accuracy of each time point to a 2 x 3 ANOVA (2 word categories by 3 conditions) to create *p*-values for main effects and interaction effects. For the cluster permutation method, we identify clusters of time where

$p < 0.05$ for at least 3 adjacent time points (main and interaction effects considered separately). For each cluster, we assign a cluster-level statistic equal to the sum of F-values for all time points within the cluster. We report the largest time cluster in time window 0-400ms and 400-650ms to account for earlier and later effects. To correct for the final cluster-level $p$-value, we permute the accuracies by randomly assigning the word category and condition labels within each participant data for 10,000 times.

**Temporal generalization matrices.** Temporal generalization matrices (TGMs; [26]) were used to test if the patterns identified with our ridge regression models were stable across time and/or contexts [26]. For simplicity, we first describe how to use a TGM to evaluate across time but within context, and then generalize to evaluating across contexts.

To evaluate across time, instead of training and testing using data from the same time window, we form a matrix M where $M_{ij}$ contains the decoding accuracy of a model trained on a window centered at time $i$ and tested on another window centered at time $j$. We leave out two averaged epochs from both time windows. We train the regression models using the N-2 remaining epochs from time window $i$, and test the regression models on the two left-out averaged epochs from time window $j$. If the neural representation of the word is consistent over time, then similar patterns will be leveraged by regression models trained on different time windows (thus yielding similar learned weights), resulting in above-chance decoding accuracy even when the train and test data are from differing time windows. If the representation of a word is stable over time, there will be high accuracy in blocks near-adjacent to the TGM diagonal, whereas if the representation re-emerges later in time there will be areas of high accuracy further from the diagonal (i.e., off-diagonal), separated from the diagonal by an area of lower accuracy.

We can also create cross-context TGMs, which test if representations of a certain word-type are consistent across contexts. Again, we form a matrix M where $M_{ij}$ contains accuracy of the prediction model trained on data from context A (e.g. phrase), time $i$ and tested on data from context B (e.g. list), time $j$. In a cross-context TGM, if the representation of a word-type is similar in both contexts at the same time points (i.e., when $i = j$), there will be high accuracy along the diagonal. If the representations are similar across the contexts, but at different times (i.e., with some lag), we see high off-diagonal accuracy. (For an excellent tutorial on TGMs, see [26]. For a more language specific interpretation, see [27]).

## Results

Trials containing behavioral errors were excluded from our analysis. Erroneous articulations included productions of wrong names and utterance repairs. Response accuracy was always above 97%. The average latency of speech onset for each context in increasing order was: adjective-only, 772ms; noun-only, 792ms; list, 897ms; and phrase, 917ms.

### Effect of category and context on word decodability (within-condition, within-timepoint decoding)

The time courses of noun and adjective decodability are shown in Fig 3, broken down by context, with dots above the x-axes indicating windows of reliable, significantly above-chance accuracy. Here, training and testing data are from the same time window and the same condition. Zero indicates the onset of the picture stimuli.

When were nouns and adjectives in general decodable above chance? While nouns were reliably decodable in all three contexts for sustained periods of time (phrase: 105–365, 385, 420-460ms; list: 110–170, 195–215, 225–355, 380ms; isolation: 140–190, 200–215, 235–240, 325-650ms), adjective decodability was mostly limited to a late time-window close to articulation in all three tasks (phrase: 595–620, 630-650ms; list: 65–245, 520–530, 565-650ms;

**A** Higher decodability of nouns over adjectives

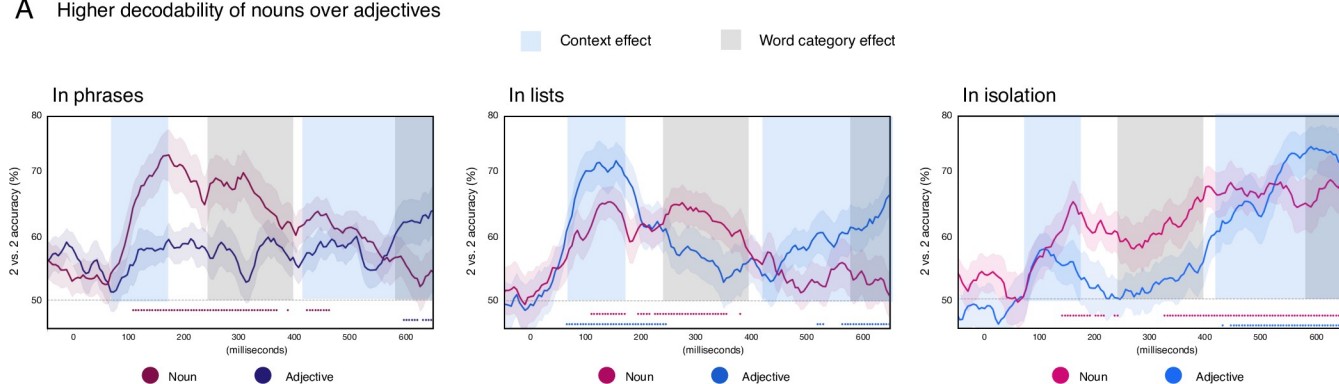

**Fig 3. Decodability across time for adjectives and nouns when presented within phrases, lists or in isolation as single words.** The grey shading indicates a significant main effect of category on decodability across all contexts, with nouns showing higher accuracy than adjectives in the mid-latency time-window of 240-395ms after picture onset. Dashed lines above the x-axes indicate when decoding accuracy was reliable for the nouns (red) and adjectives (blue). Blue shading indicates the intervals during which the main effect of context was observed, that is, higher decodability of both categories when occurring in two-word contexts (phrase or list). Though not shown, there is an interaction effect 100–190 ms.

isolation: 430, 445-650ms). In addition, adjectives in lists showed an early peak of high accuracy (65-245ms), possibly due to the somewhat artificial attention that needed to be paid to the background colors in the list task. Though we did not explicitly test for the effort needed for each naming task, we find it plausible that this may increase neural demands in some way that could also increase decodability.

In the phrase condition, only nouns showed reliable decodability, and this lasted through much of the epoch. In lists, adjectives were robustly decodable in an early time-window of ~100-250ms, while the time course of noun decodability was similar to the phrase context. Finally, isolated single words showed the same contrast as phrases: more reliable and longer lasting decodability of nouns than adjectives, though at the end of the epoch, adjective decodability did reach significance.

The effect of category and context on decodability was evaluated with a 2 x 3 ANOVA with word-category (adjective, noun) and context (isolation, list, phrase) as factors (Fig 3). A main effect of word-category was significant at 240-395ms, with higher decoding accuracies for nouns than adjectives. A main effect of word-category was also observed at 580-650ms, with higher decoding accuracies for adjectives than nouns (Fig 3, gray shading). We found an interaction effect for this 2 X 3 ANOVA at 100-190ms ($p < 0.00001$; not illustrated).

As our main question pertained to the effect of context on single word representations, we conducted further, more targeted analyses contrasting the isolated word stimuli to the phrases and list trials in two separate 2 x 2 ANOVAs. A phrasal context indeed enhanced the decodability of both nouns and adjectives, as compared to an isolated word context, but so did a list context (isolation/isolation vs phrase/phrase: context effect at 245-295ms and 510-650ms. Similarly, isolation/isolation vs list/list: context effect at 60-175ms and 405-650ms.). Thus, we are not able to attribute this increase in decodability to composition specifically. All 2 x 2 Anova results appear in the S1 Appendix.

## Generalizability of word representations across time in each context (within-condition decoding)

While the decoding analysis just described—with training and testing always using the same time point—did not reveal a compelling effect of composition on single word decodability, the analysis using TGMs did (Fig 4). In particular, during phrase planning, noun representations

Persistence and reemergence of noun representations in combinatorial contexts

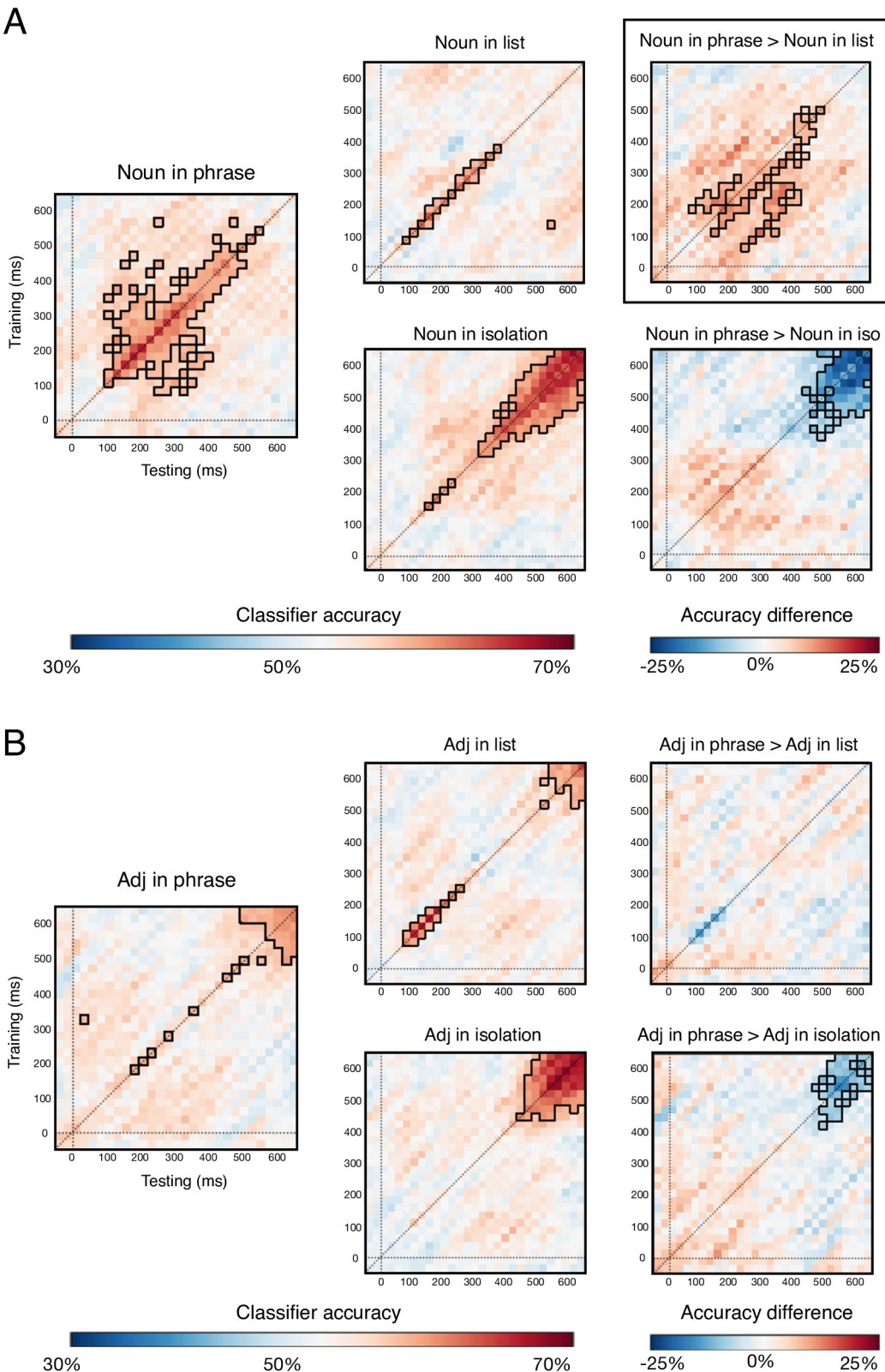

**Fig 4. Within-condition TGMs.** Within-condition TGMs showing the temporal generalizability of noun (A) and adjective (B) representations from training time X to testing time Y in the three contexts. When nouns occurred in phrases, their representations generalized between earlier and later time-points in a way that was not observed for nouns in non-phrasal contexts or for adjectives in any context. This is evidenced by the off-diagonal instances of reliable decoding in the Noun in Phrase results (A, left). The right-most column shows subtractions between phrasal and non-phrasal contexts, with black boxing indicating significant differences.

trained at early time points, starting at ~100ms, stayed active/decodable until about 400ms post picture onset (above chance accuracy regions outlined in black, Fig 4). This was not the case during the planning of lists, nor for adjective planning in any context.

## Generalizability of word representations across time and from isolation to two-word contexts (across-condition decoding)

Finally, our across-condition TGMs addressed the degree of similarity between word representations when produced as isolated words as opposed to when planned together with another word, in order to produce either a phrase or a list (Fig 5). Decoding accuracy of noun representations was reliable even when the training used isolated nouns and the testing used phrase data. The decoded representations also generalized across time, such that noun representations that were successfully decoded at 100-200ms disappeared and then re-emerged about a hundred milliseconds later, while representations at 300-400ms stayed active in a more sustained fashion until 500-600ms (above-chance regions outlined in black, Fig 5). The significantly above chance accuracy appears mostly below the diagonal, implying that the representation seen earlier in the isolation context matches the later representation in the phrase context. Isolated noun representations did not generalize well to list contexts, and did not show generalization across time (train isolation, test list in Fig 5A).

Adjective representations did not generalize from isolated contexts to phrasal contexts nearly as robustly as nouns. Mainly, evidence of shared representations across these two contexts were observed in a late time-window, close to articulation, at 500-600ms. This could reflect planning of the adjective articulation, which in all these contexts was the first word to be uttered. In a similar late time-window, isolated adjective representations generalized to adjectives in the lists, though more weakly.

## Discussion

This work addressed the nature and time course of noun and adjective representations in phrasal, isolated word, and list contexts. How does a simple combinatory context affect the neural representation of a word? Are adjectives and nouns planned in symmetric fashion during language production, or do these word types elicit different activation time courses when measured with a decoding approach? Our study yielded three major findings. First, apart from a late time window shortly prior to articulation, nouns were generally more decodable than adjectives. Second, both adjectives and nouns were more decodable when the task required the production of two words, either as a phrase or a list. And finally, we used TGMs to evaluate the temporal evolution of specific semantic representations. As these representations are activated en route from picture onset to articulation, and nouns were planned as heads of phrases, the representations active soon after picture onset stayed active up to 400ms into the epoch. Such a profile was entirely absent when nouns were planned as single words or within lists, and for all cases of adjective planning. Our across-condition decoding also provided evidence of similar representations for isolated nouns and nouns in phrases, while such evidence was much weaker for the generalizability of isolated nouns to nouns in lists, or from isolated

# A. Nouns

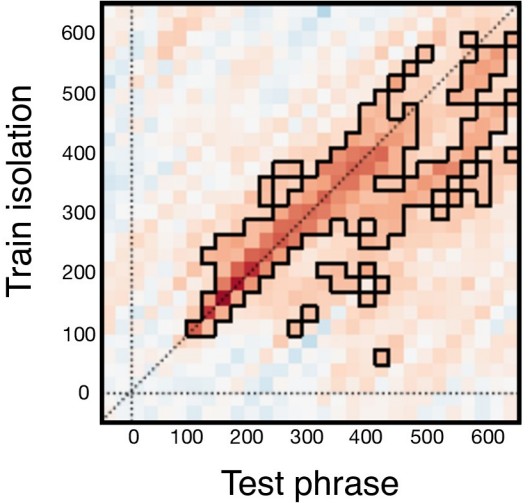
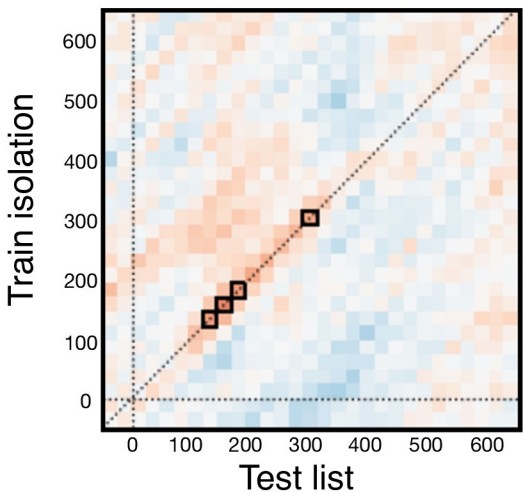

# B. Adjectives

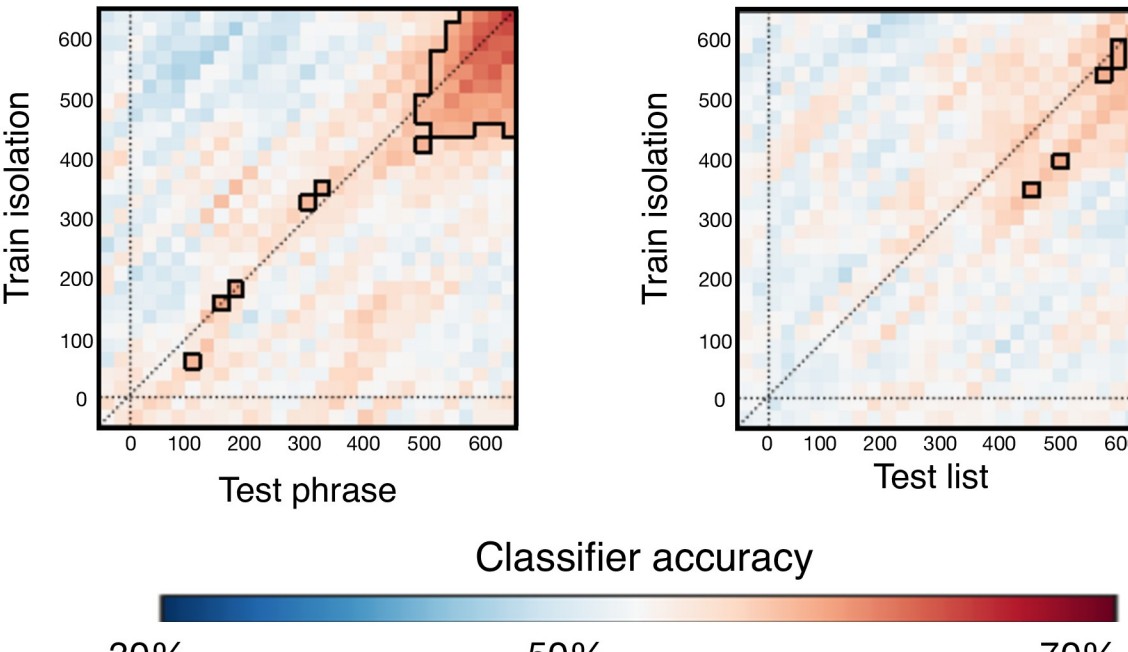

Classifier accuracy

30%          50%          70%

**Fig 5. Across-condition TGMs showing the decodability and temporal generalizability of isolated word representations to phrase and list contexts.** Classifiers were trained on nouns and adjectives as they occurred in the isolation context and then tested when those same words occurred within phrases or lists. (A) Neural representations of nouns were sufficiently similar in isolation and in phrases such that decoding was reliable starting at 100ms and lasting till almost the end of the epoch. These representations also showed temporal generalizability starting at 100ms. Representations active at 100-200ms disappeared and then re-emerged about a hundred milliseconds later, while representations at 300-400ms stayed active in a more sustained fashion until the end of the epoch. (B) Adjective representations, in contrast, did not generalize from isolated contexts to phrasal contexts nearly as robustly. Mainly, shared representations across these two contexts were observed in a late time-window, close to articulation, at 500-600ms. This could reflect planning of the adjective articulation, which was the first word to be uttered in all three depicted contexts. In a similar late time-window, isolated adjective representations generalized to adjectives in the lists, though more weakly.

adjectives to adjectives in phrases or lists. In sum, our findings suggest that during production planning, the neural representations of nouns are more stable, and therefore more decodable, than those of adjectives, and that during phrase planning, noun representations generalize across time and contexts in ways that adjective representations do not.

## Timeline of adjective and noun decodability

In general, across the full design, the first word to be uttered was decodable in a late time-window, shortly preceding speech onset. Given the late timing, the representations driving this result are likely motor related.

But earlier on 240-395ms during the language planning process, adjectives and nouns differed in their decodability. Within each context, nouns were consistently decodable, and were, for the most part, more decodable than adjectives. This was upheld by our 2 x 3 ANOVA, directly comparing adjective and noun decodability across all three contexts, which indicated greater noun than adjective decodability in early-to-mid-latency time windows. It appears, then, that noun representations stayed active for a protracted period of time, while adjective representations were only stable immediately prior to utterance onset. We also observed an effect of context, such that whenever either nouns or adjectives were planned as part of two-word utterances, whether they be lists or phrases, decoding accuracy was higher. Since we were not able to pinpoint this effect as directly relating to phrasal composition, it connects only loosely to our research question. It may stem from a higher level attention when planning two word expressions as opposed to single words. We leave this question for future work and focus our discussion on the higher decodability of nouns over adjectives.

Although color-adjectives and object-nouns differ in many ways, the restricted nature of our stimulus choices could have flattened out differences that one might observe in a more ecologically valid context. Nevertheless, a clear time-course difference was observed. With the current data, we cannot determine the cause of this, but multiple possibilities exist for future research to explore. Most interestingly, the difference in noun vs. adjective decodability could be driven by genuine semantic differences between the two word types. For example, objects are usually perceivable via multiple senses, we can feel them, see them, and perhaps hear them, but colors can be experienced only through vision. This could lead to less robust neural representations for colors. Relatedly, it has been shown that color dissociates from many other physical properties when comparing semantic representations in sighted and blind individuals [28]. Although the sighted and the blind appear to have similar representations for attributes such as shape and texture, this is not the case for color. It has been hypothesized that this may result from the lesser taxonomic value of color: color is a much weaker predictor of object kind than for example shape [28]. Thus the weaker associations between color and other object properties could also result in less detectable neural representations for color terms.

Color terms are also ambiguous in ways that we have not yet discussed [29]. For example, despite often occurring as textbook examples of a context-insensitive modifier, the interpretations of color terms are actually quite context-sensitive–compare *red hair* and *red wine* for example [30]. Although our experiment did not employ different hues, this underlying variability could nevertheless contribute to lesser decodability for color terms. There are also differences in ambiguity as regards syntactic category. Our study used two categories, nouns and adjectives. While our nouns (*bag*, *bell*, *cane*, *lamp* and *plane*) are very unlikely as adjectives in English, all our color-adjectives are actually also mass nouns (*I like milk*; *I like blue*). Given this, we cannot rule out the possibility that in the list context (red, bag), the participants were naming the background colors as nouns. If colors occurred both as nouns and as adjectives within the experiment, this also could have affected decodability.

### Role of phrasal composition in the temporal and contextual generalizability of noun representations

In addition to addressing the general time course of word representations as they participate in combinatory phrasal planning, our method allowed us to examine the relationship between representations active at different times and between representations active in different contexts. Although our within-condition, within-timepoint analysis did not reveal compelling effects of phrasal composition on either noun or adjective representations; the generalizability of noun, but not adjective, representations was clearly enhanced by a phrasal context, in the following two ways.

First, the temporal generalizability of noun representations was enhanced by a phrasal context, both as compared to isolation and list contexts (Fig 4). This finding suggests a cascade of noun representations, many of which stay active for a while. In contrast, adjective representations showed almost no generalization across time, consistent with a model in which the representation of an adjective changes consistently across time, with new representations replacing the old ones, with no sustained activations ("chain" pattern in [26]). It is interesting that the compositional context produced the most temporally generalizable representations, as the hypothesis *a priori* may have been that composition would change the representation *more* over time. However, the stimuli adjectives are largely intersective, and so it is possible that such a compositional change is less apparent in this experiment. Nouns in phrases was also the only time we observed the amount of temporal generalization reported by Fyshe et al. [2], which showed large swaths of above-chance off-diagonal accuracy. This could be for several reasons, including that the Fyshe study used a reading paradigm that displayed one word at a time, and so the representations for adjectives and nouns were neatly and predictably separated in time.

Second, the contextual generalizability of isolated nouns to two-word contexts was higher when the two words formed a phrase (Fig 5). The same was not true of adjectives, which showed much weaker decodability even within condition (Fig 4). Particularly interesting in the across-condition decoding of nouns was evidence of reactivated representations: representations active at 100-200ms disappeared and then re-emerged about a hundred milliseconds later, providing some support to the 10 Hz oscillatory activity reported by Fyshe et al. [2]. In contrast, representations decoded at 300-400ms stayed active in a more sustained fashion until the end of the epoch. Though a theoretical understanding of this detailed pattern requires further experimentation, the general finding emerging from these results is that the head of the phrase, the noun, engages a much more stable set of neural representations than its modifier, the adjective. Given our highly controlled stimulus materials and symmetric nature of the paradigm, the asymmetry is striking, and further studies could search for contexts that eliminate that asymmetry. Assessing which aspects of the decoding results stem from word order would be straightforward with a language that uses a different word order. The syntactic relation of the two elements can also be altered using a language in which noun-adjective pairs can convey a predicative relation without an overt copula: boat (is) red (cf., [31]). In sum, our findings offer a description of the representational patterns of nouns and adjectives during English phrase planning, giving rise to a host of novel hypotheses for further investigation.

## Conclusion

This work addresses the temporal evolution and context sensitivity of noun and adjective representations during phrase planning in production. We discovered a robust asymmetry between nouns and adjectives, with noun representations being generally more decodable, more consistent between isolated and phrasal contexts and more sustained over time in

phrases than those of adjectives. While our findings are not yet highly theoretically constraining, they open up a rich space of testable hypotheses about the critical factors driving the observed contrasts, in terms of either the structural or semantic properties of the two word classes.

## Supporting information

**S1 Appendix.**
(DOCX)

## Acknowledgments

We thank Chris Barker for useful discussion on the semantics of color terms and for related references.

## Author Contributions

**Conceptualization:** Maryam Honari-Jahromi, Esti Blanco-Elorrieta, Liina Pylkkänen.

**Data curation:** Maryam Honari-Jahromi, Esti Blanco-Elorrieta, Alona Fyshe.

**Formal analysis:** Maryam Honari-Jahromi, Alona Fyshe.

**Funding acquisition:** Alona Fyshe.

**Investigation:** Maryam Honari-Jahromi, Alona Fyshe.

**Methodology:** Maryam Honari-Jahromi, Liina Pylkkänen, Alona Fyshe.

**Project administration:** Liina Pylkkänen, Alona Fyshe.

**Resources:** Liina Pylkkänen, Alona Fyshe.

**Software:** Maryam Honari-Jahromi, Alona Fyshe.

**Supervision:** Brea Chouinard, Liina Pylkkänen, Alona Fyshe.

**Validation:** Maryam Honari-Jahromi.

**Visualization:** Maryam Honari-Jahromi, Esti Blanco-Elorrieta.

**Writing – original draft:** Maryam Honari-Jahromi, Brea Chouinard, Liina Pylkkänen, Alona Fyshe.

**Writing – review & editing:** Esti Blanco-Elorrieta, Liina Pylkkänen, Alona Fyshe.

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
