## [Decision Letter · Decision Letter 0]

22 Dec 2020

PONE-D-20-34763

Neural representation of words within phrases: Temporal evolution of color-adjectives and object-nouns during simple composition

PLOS ONE

Dear Dr. Fyshe,

Thank you for submitting your manuscript to PLOS ONE. After careful consideration, we feel that it has merit but does not fully meet PLOS ONE’s publication criteria as it currently stands. Both Reviewers noticed some critical aspects and required clarifications that should be addressed to improve the overall quality of the Manuscript. Therefore, we invite you to submit a revised version of the manuscript that addresses the points raised during the review process.

We look forward to receiving your revised manuscript.

Kind regards,

Nicola Molinaro, Ph.D.

Academic Editor

PLOS ONE

2.In your Data Availability statement, you have not specified where the minimal data set underlying the results described in your manuscript can be found. PLOS defines a study's minimal data set as the underlying data used to reach the conclusions drawn in the manuscript and any additional data required to replicate the reported study findings in their entirety. All PLOS journals require that the minimal data set be made fully available. For more information about our data policy, please see http://journals.plos.org/plosone/s/data-availability.

3.Thank you for stating the following in your Competing Interests section: 

"No"

Reviewers' comments:

Reviewer's Responses to Questions

**Comments to the Author**

1. Is the manuscript technically sound, and do the data support the conclusions?

Reviewer #1: Yes

Reviewer #2: Yes

2. Has the statistical analysis been performed appropriately and rigorously? 

Reviewer #1: I Don't Know

Reviewer #2: Yes

3. Have the authors made all data underlying the findings in their manuscript fully available?

Reviewer #1: Yes

Reviewer #2: Yes

4. Is the manuscript presented in an intelligible fashion and written in standard English?

Reviewer #1: Yes

Reviewer #2: Yes

5. Review Comments to the Author

Reviewer #1: The present study proposes a decoding approach to study noun adjective brain representations from MEG recordings during a production task. The original data corresponds to a previously published study in which pictures had to be described only using nouns, adjectives, or noun adjectives in a compositional or list manner.

The authors track the neural representations of noun and adjectives over time, and explore the similarity between the representation of noun and adjectives before producing the words in isolation compared to the representation of the same words produced during compositional or list like contexts by training and testing the regression models across conditions.

They conclude that noun and adjective representations behave differently: nouns are more decodable and their representation is more consistent across time and context, whereas adjectives are less decodable across contexts and time, and hence their representation is more variable.

The manuscript is well written, the objectives and hypothesis are clearly stated, the methodological approach seems correctly conducted and is consistent with the author's questions. I specially value the collaboration between research groups and the repurpose of already collected data.

Overall the paper is good but I would suggest the authors to make more explicit some specifications of the statistical analyses (see below) as well as to enrich the discussion on the following points:

Variability of noun adjective representations in comprehension and production tasks.

Invariability of noun representation across time in this production task is somewhat unexpected. In the phrasal context the combination of noun and adjective would elicit a different representation of the noun (a lamp that is red, not any lamp), and the combined representation would have to be broken down into their constituent representations (red, lamp) to produce the correct articulation. Considering that the original article shows a composition activity, noun representation when modified by the adjective should correspond to a different representation than the noun in isolation. In this sense brain representation of noun and adjective in the list context would be expected to be more consistent across time than noun in phrases.

Moreover, how does this result on adjective variability and noun robustness of representation in the production task relate to the somewhat symmetrical result in the comprehension task studied in Fyshe et al., 2019.

Finally, did the authors explore how are the decoding accuracies for nouns and adjectives when training and testing the models across the context conditions? This could provide important information on the effect of composition on noun and adjective representation.

Accuracy before speech onset

The authors mention that the increased decoding accuracy for the adjective representation prior to word articulation is motor related. This would mean that the early representation of adjectives is also partly motor related? The hypothesis suggested by the authors would benefit from some discussion on word multimodal representation at early stages of word processing.

In relation to the methods sections some points should be made more clear and detailed so reproduction is possible:

M1. The preprocessing parameters seem to be different from the original paper (i.e: epochs length, filters). If this is the case, the authors should specify all of the information concerned with the preprocessing (i.e: trial rejection, all filter specifications). If the authors did not start their study from raw data I would suggest the authors to refer the readers to the original paper for the preprocessing details. Although note that in the original paper filter specifications that would allow replicating the preprocessing are missing (filter type, cutoff frequency, filter order, roll-off or transition bandwidth, direction of computation).

M2. Authors should be more explicit on the details of the permutation cluster analysis. The null distribution was constructed taking the cluster with the maximum statistic sum? or all clusters F sum were included?.

If the statistic F corresponds to the interaction obtained by a 3x2 ANOVA for the time points 100-190ms, how were the main effect cluster times determined?, this should be more thoroughly explained and justified.

M3. I would suggest the authors to incorporate the statistical results for the 2x2 ANOVAs contrasting the isolated word stimuli to the phrases and list trials

Minor issues are detailed below separated by sections

METHODS

M4. Please specify the order of presentation for the different condition blocks

M5. “with a sampling frequency of 1000Hz” This refers to the recording parameters not the filter specifications?

M6. Note 40HZ in all caps

M7. Misphrased: total of N choose 2 pairs in p.7

M8. Authors should cite software and statistical packages used to carry the analyses, providing information on versions, etc.

RESULTS

R1. This phrase is not clear: “A main effect of word-category was significant at 240-395ms, with higher decoding accuracies for nouns than adjectives and at 580-650, with higher decoding accuracies for adjectives than nouns (Figure 2, gray shading).”

R2. As separate models for each participant were trained authors should report the variability across subjects on top of the average model performance.

R3. Misphrased: “Isolated noun representations HAD DID not generalize well to list contexts, and did not show generalization across time”

DISCUSSION

D1. Replace an for a in “of an adjectives” in p.15

D2. Misphrased “which no sustained activations” in p.15

Reviewer #2: The paper by Honari-Jahromi and colleagues presents an important and under-researched question in cognitive neuroscience of language – how stable are the neural representations of the semantic properties of adjectives and nouns across both temporal, contextual and combinatorial dimensions. To answer this question they use a MEG data and a combination of decoding techniques. Results they present are interesting and compelling. Overall I am very happy with both the quality of the paper, novelty of the question and the analysis used to answer it, however there are several areas that would require improvement and clarifications.

Major

1. The methods section describing the analysis used (p. 7) requires a lot of clarification. The stages of the decoding analysis are not clear from the text and at present it would be difficult to replicate it. For instance, what is the dimensionality of the data and predictor matrixes? What distance d was used for vector comparison (Cosine? Mahalanobis?)? In each regression (n=300) the predicted single value (the nth dimension of the semantic vector) is derived/estimated from channel x time matrix? Was the data averaged over these 100ms? Was here some dimensionality reduction performed on the sensors? Or was the stimulus value estimated by integration over all 100 time points and channels (multiplied by the learned decoder matrix)? I think for the benefit of the reader an explanatory figure (showing data dimensionality and schematic representation of steps) and formula describing the model as well as references to the exact method are necessary.

2. Semantic vectors/embeddings when derived from collocational matrixes or with neural networks to some degree reflect the frequency of the words they encode. Vectors of more frequent words tend to be more similar to each other than vectors of less frequent words. In this dataset, were the adjectives and nouns matched on word form / lemma frequency? Could better decodability of nouns across contexts and times be simply due to their higher frequency and greater vector similarity (to each other), when comped to adjectives?

3. Since decoding happened within participants (not on pooled data) and the accuracies were simply averaged, this means that decoding variability between participants was not “accounted for” and observed effects cannot be generalised outside of this sample – the models trained on one participant cannot be used to predict data in another participant/s – and this needs to be acknowledged explicitly.

Minor

1. Introduction p.3 para.2 - “complex effects of ambiguity” is a bit ambiguous. Do you mean in sentences or in narratives? In complex and naturalistic listening conditions. Please clarify.

2. Methods section p.5 “spoken and signed language as regards to the neural correlates ..” did you mean “with regards to”?

3. Did any artefact rejection or blink / heart beat removal with ICA take place? I appreciate that when using decoding artefact rejection is not strictly compulsivity as with ERPs but it is good to state this explicitly and give some references.

4. The Statistical significance section on p. 8 is somewhat difficult to read, especially the second paragraph. Was ANOVA done first across all time points and then 0.05 cutoff applied (seems that way since F values were summed for cluster mass estimate)? The word ‘label’ is ambiguous, please clarify.

5. p.9 para 2 “… similar patterns will be picked out by regression.. ” not clear, not sure ‘patterns’ is the right word. Do you mean to say regression weights learned in one window will also generate above-chance performance in another window.

6. Results p.10 was responce accuracy above 90% for all participants?

7. Figure 2 – most of the figure 1 caption text belongs in the text of the results section, not in the figure caption. Also please explain exactly why you think the demand to attend to the background colour improved decodability for adjectives in lists.

8. p.11 “we found an interaction effect...” please explain

9. Figure 3 and 4 – does the black contouring indicate statistical significance? Again, please consider putting text that describes results out of the caption and into main text.

10. It would be useful throughout the manuscript to refer to not simply ‘representations’ but semantic or lexicon-semantics representation, since this is what model was trained to decode (as opposed to say phonological representations).

---

## [Author Response · Author response to Decision Letter 0]

9 Feb 2021

We thank our reviewers for their thoughtful comments on our work. We have addressed them in the new revision, and discuss each comment below. Our response is also provided as a color-coded pdf, which is likely easier to read than what follows here.

Reviewer #1: 

Variability of noun adjective representations in comprehension and production tasks.

*Invariability of noun representation across time in this production task is somewhat unexpected. In the phrasal context the combination of noun and adjective would elicit a different representation of the noun (a lamp that is red, not any lamp), and the combined representation would have to be broken down into their constituent representations (red, lamp) to produce the correct articulation. Considering that the original article shows a composition activity, noun representation when modified by the adjective should correspond to a different representation than the noun in isolation. In this sense brain representation of noun and adjective in the list context would be expected to be more consistent across time than noun in phrases. 

We agree that under composition meaning changes. However, color adjectives are largely intersectional and in this case, would not be expected to have a large effect on the semantics of the noun. In addition, our analyses search for the part of the representation that stays the same. There could be additional brain activity associated with the adjective, and so long as it doesn't change or otherwise hinder the representation of the noun we would expect to see no difference to the decoding accuracy of the noun. We have updated the discussion (section “Role of Phrasal Composition in the Temporal and Contextual Generalizability of Noun Representations”) to reflect this.

Interestingly there is little variability for nouns in lists (was Figure 3A, now Fig 4A, Noun list TGM). We believe the absence of off-diagonal above-chance accuracy stems from the fact that the noun need not be maintained in the mental workspace to be composed. Rather, it can be stored away until articulation is imminent. 

Moreover, how does this result on adjective variability and noun robustness of representation in the production task relate to the somewhat symmetrical result in the comprehension task studied in Fyshe et al., 2019. 

We have added a discussion of this to the section “Role of Phrasal Composition in the Temporal and Contextual Generalizability of Noun Representations”

Finally, did the authors explore how are the decoding accuracies for nouns and adjectives when training and testing the models across the context conditions? This could provide important information on the effect of composition on noun and adjective representation. 

In the paper we included what we felt were the best cross-condition tests of how the meaning of isolated words change as they are used in phrases. Figure 5 (was Figure 4) shows training in isolation and testing in either list or phrase, and shows that the isolated representations are quite similar to the phrasal representation, but not to the list representations. 

Although this was not our primary question of interest, we had run list/phrase ANOVAs, which indicated a consistent effect of category (nouns more decodable than adjectives mid-epoch, and adjectives more decodable than nouns late epoch) with no interaction when comparing list/phrase to phrase/phrase, but an interaction when comparing list/list to list/phrase, supporting our within-context word category effects and our conclusion that phrasal context enhances decodability of nouns.

Accuracy before speech onset

The authors mention that the increased decoding accuracy for the adjective representation prior to word articulation is motor related. This would mean that the early representation of adjectives is also partly motor related? The hypothesis suggested by the authors would benefit from some discussion on word multimodal representation at early stages of word processing.

We do not think the early adjective signal is motor related, because there is no off-diagonal accuracy in any graph that connects a late (pre-utterance) period to an early period.

In relation to the methods sections some points should be made more clear and detailed so reproduction is possible:

M1. The preprocessing parameters seem to be different from the original paper (i.e: epochs length, filters). If this is the case, the authors should specify all of the information concerned with the preprocessing (i.e: trial rejection, all filter specifications). If the authors did not start their study from raw data I would suggest the authors to refer the readers to the original paper for the preprocessing details. Although note that in the original paper filter specifications that would allow replicating the preprocessing are missing (filter type, cutoff frequency, filter order, roll-off or transition bandwidth, direction of computation). 

We have updated the text to more clearly state the differences between processing:

“Trials were epoched at 100ms before to 700ms after stimuli onset to avoid contamination via motion artifact coinciding with overt speech and noise was reduced using Continuously Adjusted Least-Squares Method (Adachi et al., 2001). Epochs were baseline corrected using the average of a 100ms interval prior to the stimulus onset. Unlike Blanco-Elorrieta et al., we rejected only those trials with erroneous responses. MEG signals were band-passed using a Butterworth filter of order 20 between 0.1Hz and 40Hz. We used no ICA artefact rejection or blink / heart beat removal, as such artefacts are less problematic in decoding studies.”

M2. Authors should be more explicit on the details of the permutation cluster analysis. The null distribution was constructed taking the cluster with the maximum statistic sum? or all clusters F sum were included?

If the statistic F corresponds to the interaction obtained by a 3x2 ANOVA for the time points 100-190ms, how were the main effect cluster times determined?, this should be more thoroughly explained and justified. 

The nonparametric clustering permutation based on 2by3 ANOVAs is done as follows. No model retraining is involved. The procedure is as follows:

1. run a 2by3 ANOVA for every time point and obtain "f-ratio "and "p_value" for each effect ( word category (2), condition (3) and interaction)

2. find clusters that span longer than 15ms (3 consecutive time points) where p-value of each point is less than 0.05 , record sum(f-ratios) for each cluster

3. assign "cluster corrected p-value" based on sum(f-ratios) from the null distribution formed as below:

 repeat 10*1000{

1. shuffle the 2by3 matrix (full shuffle not row-wise or column-wise)

2. calculate 2by3 ANOVA as before

3. for the above clusters, record sum(f-ratios)

}

4. report the largest cluster in 0-400 and 400-650 to account for early and late effects.

We have incorporated this description into the “Statistical Significance” section.

M3. I would suggest the authors to incorporate the statistical results for the 2x2 ANOVAs contrasting the isolated word stimuli to the phrases and list trials 

We have incorporated the 2 x 2 into an appendix to the paper. Here are the results from those tests:

2 x 2 ANOVA comparing isolation/isolation against phrase/phrase: 

Main effect of context at 245-295ms and 510-650ms; 

Main effect of word-category at 120-355ms; 

no interaction effect. 

2 x 2 ANOVA comparing list/list against phrase/phrase: 

main effect of context at 70-100ms and 440-470ms; 

main effect of word-category at 250-350ms and 605-650ms; 

interaction effect at 95-195ms 

2 x 2 ANOVA comparing isolation/isolation against list/list: 

main effect of context at 60-175ms and 405-650ms; 

main effect of word-category at 290-360ms; 

interaction effect at 145-190ms

Minor issues are detailed below separated by sections

METHODS

M4. Please specify the order of presentation for the different condition blocks. 

The order of blocks was randomized across participants with the only constraint that two blocks of the same condition never appeared consecutively. We have clarified this in the paper.

M5. “with a sampling frequency of 1000Hz” This refers to the recording parameters not the filter specifications?

fixed

M6. Note 40HZ in all caps

fixed

M7. Misphrased: total of N choose 2 pairs in p.7

This is terminology typically used for binomial coefficients. For clarity, we included the mathematical notation with the exact number. We also mentioned the excluded test pairs in the manuscript. Epoch averaging yielded 4 averaged epochs per noun (5 nouns) and 20 averaged epochs in total. Due to repeated nouns among epochs, we have excluded test pairs with the same nouns in our analysis. There are 30 of such pairs. Same procedure followed for adjectives. 

M8. Authors should cite software and statistical packages used to carry the analyses, providing information on versions, etc. 

Decoding analysis was done in scikit-learn (Pedregosa et al. 2011). Statistical significance test was done in MNE-Python (Gramfort et al. 2013) and FieldTrip (Oostenveld et al. 2011). The code for all analysis and respective software versions is available at https://github.com/mahon94/compositionInBrain.

Citations have been added to the draft.

RESULTS

R1. This phrase is not clear: “A main effect of word-category was significant at 240-395ms, with higher decoding accuracies for nouns than adjectives and at 580-650, with higher decoding accuracies for adjectives than nouns (Figure 2, gray shading).”

Rephrased

R2. As separate models for each participant were trained authors should report the variability across subjects on top of the average model performance. 

SEM appears as the shaded area surrounding the lines in our line charts (Figure 3, was Figure 2).

R3. Misphrased: “Isolated noun representations HAD DID not generalize well to list contexts, and did not show generalization across time”

Fixed

DISCUSSION

D1. Replace an for a in “of an adjectives” in p.15

Fixed

D2. Misphrased “which no sustained activations” in p.15

Fixed

Reviewer #2: The paper by Honari-Jahromi and colleagues presents an important and under-researched question in cognitive neuroscience of language – how stable are the neural representations of the semantic properties of adjectives and nouns across both temporal, contextual and combinatorial dimensions. To answer this question they use a MEG data and a combination of decoding techniques. Results they present are interesting and compelling. Overall I am very happy with both the quality of the paper, novelty of the question and the analysis used to answer it, however there are several areas that would require improvement and clarifications.

Major

1. The methods section describing the analysis used (p. 7) requires a lot of clarification. The stages of the decoding analysis are not clear from the text and at present it would be difficult to replicate it. For instance, what is the dimensionality of the data and predictor matrixes? What distance d was used for vector comparison (Cosine? Mahalanobis?)? In each regression (n=300) the predicted single value (the nth dimension of the semantic vector) is derived/estimated from channel x time matrix? Was the data averaged over these 100ms? Was here some dimensionality reduction performed on the sensors? Or was the stimulus value estimated by integration over all 100 time points and channels (multiplied by the learned decoder matrix)? I think for the benefit of the reader an explanatory figure (showing data dimensionality and schematic representation of steps) and formula describing the model as well as references to the exact method are necessary. 

Details of this analysis are added to the paper. We train d independent ridge regression models to predict each dimension of a semantic space Y∈RN×dfrom the MEG dataset X∈RN×p where d=300 is dimensions of the Skip-gram vectors,N=20 is the total number of averaged epochs which are reshaped to vector of length p=c×t, for c=208 MEG gradiometer sensors with t=100 time samples. In each regression (n=300) the predicted single value (the nth dimension of the semantic vector) is estimated from the reshaped average epoch vector. The averaging of epochs is discussed in the preprocessing section. While we did not use any dimensionality reduction method, we use a well-known kernel method based on singular value decomposition to speed up training and regularized regression to prevent overfitting. As mentioned in the draft, we used cosine distance for 2-vs-2 tests. We have added a new figure (Figure 2) to help clarify these points.

2. Semantic vectors/embeddings when derived from collocational matrixes or with neural networks to some degree reflect the frequency of the words they encode. Vectors of more frequent words tend to be more similar to each other than vectors of less frequent words. In this dataset, were the adjectives and nouns matched on word form / lemma frequency? Could better decodability of nouns across contexts and times be simply due to their higher frequency and greater vector similarity (to each other), when comped to adjectives? 

We controlled for frequency across word types. English frequency was extracted from Balota et al., 2007. Noun freq. mean = 14396 (sd = 8242); Adj freq. Mean = ; 14976 (t= -0.47, p = .640742).

Balota, D. A., Yap, M. J., Hutchison, K. A., Cortese, M. J., Kessler, B., Loftis, B., ... & Treiman, R. (2007). The English lexicon project. Behavior research methods, 39(3), 445-459.

We have added the clarification and citation to the paper.

3. Since decoding happened within participants (not on pooled data) and the accuracies were simply averaged, this means that decoding variability between participants was not “accounted for” and observed effects cannot be generalised outside of this sample – the models trained on one participant cannot be used to predict data in another participant/s – and this needs to be acknowledged explicitly.

Thank you for raising this point. We added this to the methodology section:

It should be noted that every model was trained separately for each participant. Thus, the patterns underlying the decoding accuracies we observed may not be stable across people, and our analysis did not test for such stability. Rather, we tested for the presence of a pattern, and if the patterns generalize across time and condition within a participant’s data. Our methodology then tests if the average decoding accuracy (a function of the participant-specific patterns) shows stable patterns across participants.

Minor

1. Introduction p.3 para.2 - “complex effects of ambiguity” is a bit ambiguous. Do you mean in sentences or in narratives? In complex and naturalistic listening conditions. Please clarify. 

The intention was to refer back to the types of cases mentioned in the first paragraph. This has now been clarified. 

Revision: “thus did not investigate ambiguous cases such as those just mentioned.”

2. Methods section p.5 “spoken and signed language as regards to the neural correlates ..” did you mean “with regards to”? 

Fixed

3. Did any artefact rejection or blink / heart beat removal with ICA take place? I appreciate that when using decoding artefact rejection is not strictly compulsivity as with ERPs but it is good to state this explicitly and give some references. 

We did not apply ICA, and updated the methodology to make this explicit

4. The Statistical significance section on p. 8 is somewhat difficult to read, especially the second paragraph. Was ANOVA done first across all time points and then 0.05 cutoff applied (seems that way since F values were summed for cluster mass estimate)? The word ‘label’ is ambiguous, please clarify. 

This section was unclear. We’ve reworked it, and hope it is clearer now. We have two completely separate statistical tests:

1. permutation tests to determine if accuracy is above chance 

2. anova test to find condition/word-category effects: we did not retrain any models for this. We permuted 2-vs-2 accuracies by randomly assigning word categories or condition labels.

5. p.9 para 2 “… similar patterns will be picked out by regression.. ” not clear, not sure ‘patterns’ is the right word. Do you mean to say regression weights learned in one window will also generate above-chance performance in another window.

Reworded

6. Results p.10 was responce accuracy above 90% for all participants? 

Yes.

7. Figure 2 – most of the figure 1 caption text belongs in the text of the results section, not in the figure caption. Also please explain exactly why you think the demand to attend to the background colour improved decodability for adjectives in lists.

Thank you for pointing this out. Much of this text has been moved to the results section. 

Naming the background color intuitively feels more effortful and less natural than naming the object-color. We find it plausible that this may increase neural demands in some way that could also increase decodability. It is a speculation of course, but we want to offer it as a possible way to think about this adjective effect. This note has now been added on p. 11. 

8. p.11 “we found an interaction effect...” please explain

Clarified to which ANOVA the interaction belongs.

9. Figure 3 and 4 – does the black contouring indicate statistical significance? Again, please consider putting text that describes results out of the caption and into main text.

We have incorporated mention of the black outlines into the main text.

10. It would be useful throughout the manuscript to refer to not simply ‘representations’ but semantic or lexicon-semantics representation, since this is what model was trained to decode (as opposed to say phonological representations). 

Changes made to the introduction, and throughout where needed.

---

## [Editor Report · Decision Letter 1]

12 Feb 2021

Neural representation of words within phrases: Temporal evolution of color-adjectives and object-nouns during simple composition

PONE-D-20-34763R1

Dear Dr. Fyshe,

We’re pleased to inform you that your manuscript has been judged scientifically suitable for publication and will be formally accepted for publication once it meets all outstanding technical requirements.

Kind regards,

Nicola Molinaro, Ph.D.

Academic Editor

PLOS ONE

---

## [Editor Report · Acceptance letter]

22 Feb 2021

PONE-D-20-34763R1 

Neural representation of words within phrases:Temporal evolution of color-adjectives and object-nouns during simple composition 

Dear Dr. Fyshe:

I'm pleased to inform you that your manuscript has been deemed suitable for publication in PLOS ONE. Congratulations! Your manuscript is now with our production department. 

Kind regards, 

on behalf of

Dr. Nicola Molinaro 

Academic Editor

PLOS ONE